The biogeography of bent-toed geckos, Cyrtodactylus (Squamata: Gekkonidae)

Grismer L. Lee lgrismer@lasierra.edu 1
Poyarkov Nikolay A. n.poyarkov@gmail.com 2 3
Quah Evan S.H. 1 4
Grismer Jesse L. 1
Wood Jr Perry L. 5 6
1 Department of Biology, La Sierra University , Riverside , CA , United States of America
2 Faculty of Biology, Department of Vertebrate Zoology, Moscow State University , Moscow , Russia
3 Joint Russian-Vietnamese Tropical Research and Technological Center , Hanoi , Vietnam
4 Institute for Tropical Biology and Conservation, Universiti Malaysia Sabah, Jalan UMS , Kota Kinabalu , Sabah Malaysia
5 Department of Biological Sciences & Museum of Natural History, Auburn University , Auburn , AL , United States of America
6 Department of Ecology and Evolutionary Biology, University of Michigan , Ann Arbor , MI , United States of America
Measey John
Electronic publication date: 2022 Mar 22
Publication date: 2022
Volume: 10
Electronic Location ID: e13153
Received 2021 Nov 30; Accepted 2022 Mar 1
Copyright: ©2022 Grismer et al.
Copyright year: 2022
Copyright holder: Grismer et al.
License: This is an open access article distributed under the terms of the Creative Commons Attribution License, which permits unrestricted use, distribution, reproduction and adaptation in any medium and for any purpose provided that it is properly attributed. For attribution, the original author(s), title, publication source (PeerJ) and either DOI or URL of the article must be cited.
License URL: https://creativecommons.org/licenses/by/4.0/

Keywords: South Asia, Indochina, Southeast Asia, Melanesia, Phylogeny, Centers of origin, Dispersal, Sundaland, Wallacea

Funding: Russian Science Foundation 19-14-00050 Nikolay A. Poyarkov’s work was supported by the Russian Science Foundation (RSF grant No. 19-14-00050; specimen collection, molecular, phylogenetic and morphological analyses, data analysis). The funders had no role in study design, data collection and analysis, decision to publish, or preparation of the manuscript.

==============================
The gekkonid genus Cyrtodactylus is the third largest vertebrate genus on the planet with well over 300 species that range across at least eight biogeographic regions from South Asia to Melanesia. The ecological and morphological plasticity within the genus, has contributed to its ability to disperse across ephemeral seaways, river systems, basins, land bridges, and mountain ranges—followed by in situ diversification within specific geographic areas. Ancestral ranges were reconstructed on a mitochondrial phylogeny with 346 described and undescribed species from which it was inferred that Cyrtodactylus evolved in a proto-Himalaya region during the early Eocene. From there, it dispersed to what is currently Indoburma and Indochina during the mid-Eocene—the latter becoming the first major center of origin for the remainder of the genus that seeded dispersals to the Indian subcontinent, Papua, and Sundaland. Sundaland became a second major center of radiation during the Oligocene and gave rise to a large number of species that radiated further within Sundaland and dispersed to Wallacea, the Philippines, and back to Indochina. One Papuan lineage dispersed west to recolonize and radiate in Sundaland. Currently, Indochina and Sundaland still harbor the vast majority of species of Cyrtodactylus.

Introduction

The gekkonid genus Cyrtodactylus is the third largest vertebrate genus on the planet (following the frog genus Pristimantis and the lizard genus Anolis) with well over 300 species whose extensive distribution occupies at least eight biogeographic regions and crosses a number of well-established current biogeographic barriers from South Asia to Melanesia (Grismer et al., 2021a; Uetz, Freed & Hošek, 2021). Recent studies concerning habitat preference within Cyrtodactylus (Grismer et al., 2020; Grismer et al., 2021a), underscore the evolution of its broad ecological diversity and plasticity which likely has contributed to its taxonomic diversity and vast distribution. The phylogenetic relationships and geographic distribution among, and within, its 32 species groups, are indicative of its remarkable dispersal capabilities in being able to cross ephemeral seaways, major river systems, basins, mountain ranges, and land bridges—followed by extensive in situ diversification within specific geographic areas (Grismer et al., 2021a; Grismer et al., 2021b). The fact that this widely distributed, ecologically plastic, and hyper-diverse genus extends across what arguably has been one of the most tectonically active regions of the globe, provides an ideal opportunity to uncover the likely geological and ecological events that may have contributed to its phylogenetic structure and diversity. Furthermore, just such an analysis can offer insights into what may have been the dispersal patterns of colonization that contributed to a multitude of in situ radiations as Cyrtodactylus spread across continental Asia, the Philippines, and the entire Indo-Australian Archipelago.

Although there have been several localized historical biogeographic studies focused on different monophyletic groups of Cyrtodactylus (e.g., Agarwal et al., 2014; Grismer et al., 2014; Grismer et al., 2018; Agarwal & Karanth, 2015; Nguyen et al., 2017; O’Connell et al., 2019; Grismer & Davis, 2018; Siler et al., 2010; Welton et al., 2010a; Welton et al., 2010b; Oliver et al., 2018; Davis et al., 2020), their inherently narrow geographic perspective leaves them somewhat limited in scope with respect to the biogeography of the genus as a whole. Wood Jr et al. (2012), provided the only genus-wide analysis that included 68 species that represented all the major clades of Cyrtodactylus from across its range. They advanced a number of broad scenarios and hypotheses within a well-supported phylogenetic framework to account for the divergence of many of their recovered clades within certain biogeographic regions—many of which are supported here (see below). However, there was little information on how tectonic events or paleoecology may have contributed to these patterns. Here, we revisit their foundational work using additional tectonic and paleoecological data concerning the position of the Indian subcontinent prior to its collision with continental Asia (Acton, 1999; Aitchison, Ali & Davis, 2007; Köhler & Glaubrecht, 2007; Ali & Aitchison, 2008; Aitchison & Ali, 2012; Ding et al., 2017), the uplift of the Himalayas and the Tibetan Plateau (Farve et al., 2015), orogeny across northwestern Indochina (Feton et al., 1997; Fenton, Charusiri & Wood, 2003; Upton, Bristow & Hurford, 1995; Upton et al., 1997), and geographic reconstructions of southern Indochina and the Indo-Australian Archipelago from 60 million years ago (mya) to present (Hall, 1998; Hall, 2012; Hall, 2013; Cao et al., 2017). We do so in the context of a well-supported mitochondrial gene tree comprised of 346 described and undescribed species that encompass the entire range of the genus and its 32 monophyletic species groups (Grismer et al., 2021a; Grismer et al., 2021b).

Materials & Methods

The BEAST (Bayesian Evolutionary Analysis by Sampling Trees (BEAST)) used in Grismer et al. (2021b) is used here and was used to estimate the ancestral range at each node using the R package BioGeoBEARS (Matzke, 2013; Matzke, 2014) as in Grismer et al. (2017). Tree construction follows Grismer et al. (2021b). The tree file, nexus file and geographic distribution files are in Supplemental Information. All GenBank Data and accompanying references are in the Supplemental Information of Grismer et al. (2021b). The only additional GenBank numbers are MW111438 for Cyrtodactylus tibetanus (YPX1413), MW111425 for C. zhaoermi (YPX1433), and OK626314 for C. cf. brevipalmtus (USMHC 2555).

Biogeographic regions (Fig. 1)

A presence or absence species geography file was constructed using generally the same biogeographic areas employed by Wood Jr et al. (2012) with modifications and further refinement herein (see below). These areas were not randomly chosen but are regions among which cladogenic turnover within a broad range of taxonomic groups has been demonstrated. No species ranged outside their designated biogeographic region and as such was allowed to occupy only a single area.

Figure 1 The timing of early colonization routes.

Extinct, ancestral taxa are designated as A# and enclosed within rounded rectangles. Arrows originating from a rectangle represent the divergence of an ancestor (A) and its subsequent radiation within the same region and/or colonization of a different region. Arrows originating from a common base represent a single divergence event and the formation of sister lineages. Arrows not sharing a common base (e.g. those originating in A7) represent independent divergence events at different periods in time within that region (e.g. Sundaland). Numbers at the base of all arrows are approximate mean divergence times in millions of years. Base Map created using simplemappr.net. Photo by L. Lee Grismer.

Himalaya region

This region extends from eastern Pakistan to northern Myanmar across the Tibetan Plateau and trans-Himalayan landscapes (sec. Agarwal et al., 2014) south of the Indus and Brahmaputra Rivers, to the southern foothills of the Himalayan Mountains north of the Indo-Gangetic Plain. This is the combined areas of the trans-Himalayas, western Himalayas, and eastern Himalayas of Agarwal et al. (2014) and encompasses the Tibet region of Wood Jr et al. (2012). This region is consistent with the Himalayan region of Xu et al. (2020) except for the inclusion of the Tibetan Plateau.

India-Sri Lanka region

This region is that portion of the Indian subcontinent south of the Indo-Gangetic Plain and Ganga River from the Gulf of Kutch in the west through the province of Odisha in the east. This region includes the Island of Sri Lanka. Grismer & Davis (2018) considered this region as part of Indoburma. This is the same as the South Asia region of Xu et al. (2020).

Indoburma region

Based on their phylogeny, Wood Jr et al. (2012) considered Indochina to be comprised of three distinct regions: Western, Central, and Eastern Indochina. The Indoburma Region used herein, with slight modifications, is equivalent to their West Indochina region. This region includes the eastern section of Indo-Gangetic Plain and southern foothills of the eastern Himalayas and extends east through Bangladesh to the eastern edge of the Ayeyarwady Basin at the foot of the Shan Plateau in central Myanmar. From the eastern Indo-Gangetic Plain northeast of the Ganga River, it extends south to the Bay of Bengal east of the Padma River. Wood Jr et al. (2012) considered the eastern edge this region to include the Salween Basin and the western portion of the Shan Plateau—the latter comprising the Tenasserim Mountains and Thai Highlands (Barr & Macdonald, 1991; Sone & Metcalfe, 2008) west of the Salween River. However, the Salween Basin and these upland areas lie to the east of the Three Pagodas fault in the south and the Sagaing fault in the north (Morley, Charusiri & Watkinson, 2011) and species within these regions are closely related to species from the Indochina Region (Grismer & Davis, 2018). In a more fine-scaled analysis, Grismer & Davis (2018) designated this narrow region as the eastern Myanmar region. Herein, we restrict the eastern boundary of the Indoburma region to the west so as to exclude the Salween Basin and the entire Shan Plateau.

Indochina region

The Indochina region as used herein, comprises the Central and East Indochina regions of Wood Jr et al. (2012) and the eastern Myanmar and Indochina regions of Grismer & Davis (2018). It extends from the western edge of the Shan Plateau of eastern Myanmar, southeastward across tropical Asia to the South China Sea along coastal Vietnam, and southward from the Red River Valley in northern Vietnam and the Yunnan Province of China, to an area on the Burma-Thai-Malay Peninsula generally referred to as the Isthmus of Kra. We here demarcate this boundary along the Surat-Thani Line along the Khlong Marul Fault (Watkinson et al., 2011; Poyarkov et al., 2021). We combined the Central and East Indochina regions of Wood Jr et al. (2012) into one group given that the phylogenetic analyses herein indicate that a number of species groups have taxa that occur within both these designated regions. However, this is not to say there is no phylogenetic substructuring between these regions, only that it is not as clear as previously proposed by Wood Jr et al. (2012) with their more limited sampling. Combining these regions herein resulted is less ambiguous reconstructions of ancestral areas. More focused biogeographic analyses of the monophyletic lineages within the Indochina region should, however, incorporate their more fine-grained geographic partitioning.

Sundaland region

This region extends from the Andaman and Nicobar Islands and the Isthmus of Kra in the northwest, southeastward through the Indo-Australian Archipelago to Wallace’s Line, including Christmas Island in the India Ocean and the Anamba and Natuna archipelagos in the South China Sea. This is the same region as the Sunda portion of the Sunda/Wallacea region of Wood Jr et al. (2012).

Philippine region

This region includes the entire Philippine Archipelago including Palawan and Balabac islands along with their smaller associated islands. This is same region used by Wood Jr et al. (2012).

Wallacea region

This region includes all the islands between Wallace’s and Lydekker’s Lines (sec. Ali, Aitchison & Meiri, 2020; Kreft & Jetz, 2010, respectively). It also includes East Montalivet Island off the northwest coast of Australia even though it occurs south of Lydekker’s Line. East Montalivet Island is the type locality of Cyrtodactylus kimberleyensis which is deeply nested within a Wallacean clade of species (Grismer et al., 2021b). This is the same region as the Wallacea portion of the Sunda/Wallacea region of Wood Jr et al. (2012). The nomeclatural history of these lines is reviewed by Ali & Heaney (2021). We retain their original names.

Papua region

The Papua region encompasses those islands on the Sahul Shelf that include New Guinea, Indonesian islands east of Lydekker’s Line, and the islands of western Melanesia as far east as the Solomon Islands, Bougainnville Island, and the Bismarck Archipelago. We also include northeast Queensland, Australia as this area encompasses the locality for Cyrtodactylus hoskini and C. mcdonaldi which are nested within a Papuan clade of species (Grismer et al., 2021b). This is the same Papuan region of Wood Jr et al. (2012). This is the same as the Australia region of Xu et al. (2020).

Results

The time-calibrated BEAST analysis recovered a phylogeny with well-supported nodes (Bayesian Posterior Probabilities (BPP) ≥ 90) throughout the tree and matches phylogenies generated from multilocus data (Wood Jr et al., 2012; Agarwal et al., 2014; Grismer et al., 2021a; Fig. 2). The phylogeny indicates that Cyrtodactylus diverged from its sister lineage Hemidactylus (Gamble et al., 2012) during the early Eocene at approximately 55 mya and continued to radiate across Asia up until the Pleistocene (Figs. 2 and 3). Diversification of the major lineages (i.e., most of the species groups and the Melanesian and Indochinese clades) happened between approximately 44–33 mya with the additional species groups evolving up until 22 mya (Fig. 3)

Figure 2 Time calibrated BEAST phylogeny.

Bayesian posterior probabilities (BPP), mean divergence times, and 95% highest posterior densities (HPD) in millions of years are shown.

Figure 3 Dispersal–Extinction–Cladogenesis +J (DEC + J) chronogram.

Annotations at the nodes (A1–A24) represent ancestral taxa (A). Biogeographic regions are color-coded in the upper left and are depicted in Fig. 1. Photo by L. Lee Grismer.

The BioGeoBEARS model comparisons show that DEC +J model is the best fit to the data and most likely to infer the correct ancestral range at each node being that it had the lowest AIC and AIC-wt scores (Table 1). Despite criticisms of the +J parameter (Ree & Sanmartín, 2018), it is noteworthy that all the trees generated in all the analyses generally recovered the same ancestral range for each node, thus converging on the same biogeographical scenario.

Table 1 Model testing for the BioGeoBEARS analysis with and without found-event speciation (+J).

Model	LnL	Number of parameters	d	e	J	AIC	AIC-wt	
DEC	−119.2295	2	2.70E−04	1.00E−12	0.00000000	242.4590	2.97E−06	
DEC+J	−105.5017	3	3.44E−05	1.00E−12	0.002350594	217.0033	1.00E+00	
DIVELIKE	−156.3889	2	5.85E−04	1.00E−12	0.00000000	316.7778	2.16E−22	
DIVELIKE+J	−127.0716	3	1.18E−10	1.00E−12	0.004438113	260.1432	4.29E−10	
BAYARALIKE	−230.9748	2	5.05E−04	6.14E−02	0.00000000	465.9496	8.75E−55	
BAYARALIKE+J	−127.0716	3	1.18E−10	1.00E−12	0.004438113	260.1432	4.29E−10	
Notes.

Models tested: dispersal-extinction-cladogenesis (DEC); Bayesian analysis of biogeography when the number of areas is large (BayArea); and dispersal-vicariance (DIVA).

d rate of dispersal

e rate of extinction

Discussion

Overview of regional dispersal patterns of Cyrtodactylus (Fig. 4)

Cyrtodactylus evolved during the early Eocene approximately 52 million years ago (mya) and gave rise to two small monophyletic Himalayan lineages before dispersing into what is currently Indoburma and Indochina. This is consistent with the step-wise scenario proposed for the Himalayan region by Xu et al. (2020). The colonization of Indoburma resulted in the evolution of a modest number of species but that of Indochina seeded the origin and evolution of the remainder of the genus from India and Sri Lanka to Melanesia (see alsoWood Jr et al., 2012 and Xu et al. (2020). Three independent, consecutive, episodes of dispersal out of Indochina gave rise to radiations in the regions of Sundaland, India-Sri Lanka, and Papua. The Sundaland radiation spread eastward and gave rise to groups in Wallacea on at least three separate occasions with dispersals back and forth between the two regions for approximately 15 million years, during which there was also a dispersal to Papua from Wallacea. The Sundaland radiation also gave rise to a Philippine clade and a single dispersal back to Indochina with a subsequent radiation therein. The India-Sri Lanka radiation spread across southern and central India and onto Sri Lanka. The Paupa radiation gave rise to several small, but distinctive clades throughout that region, as well as one clade that dispersed back west to colonize and radiate in Sundaland. Currently, distantly related and independently evolved clades throughout Indochina and Sundaland account for the majority of the diversity within Cyrtodactylus and continue to grow at an ever-increasing rate with ongoing discoveries of new species (see Uetz, Freed & Hošek, 2021).

Evolution, dispersal, and colonization of Cyrtodactylus (Figs. 1 and 3)

Below we discuss the biogeographic patterns of the major clades and species groups recently delimited in Cyrtodactylus (Grismer et al., 2021a; Grismer et al., 2021b). We do not discuss many of the fine-scaled biogeographic patterns within these clades and species groups that are detailed elsewhere, but instead, direct the reader to these fine publications at the relevant junctures. All ages presented are mean values and the highest posterior densities (HPD) about the means are presented in Fig. 2. Wood Jr et al. (2012) and Agarwal et al. (2014) noted that a number of cladogenic events among basal lineages of Cyrtodactylus and other gekkonid genera occurred early on the southern edge of the Asian continent in a “proto-Himalayan” region, or near what is currently the Tibetan Plateau from ∼62–52 million years ago (mya). These divergences were hypothesized to result from changing landscapes during ongoing compression of the Neotethyan island arcs, including the Kohistan Arc—situated between the Indian subcontinent and Eurasia (Ali & Aitchison, 2008; Morley, 2018)—as the Indian subcontinent began approaching continental Asia (Fig. 5A). These events are proposed to have sequentially given rise to the ancestor of the tibetanus group at ∼51.6 mya, and the ancestor of the lawderanus group, and ancestor (A)1 ∼47.7 mya (Fig. 3). The dates here are slightly younger (∼5 million years) than those of Agarwal et al. (2014) but are commensurate with those of Wood Jr et al. (2012) upon which our tree was calibrated.

Figure 4 Overview and timing of major dispersal patterns.

Numbers are estimated node ages in millions of years from the BEAST time-calibrated tree. Photo by L. Lee Grismer.

Figure 5 Paleogeographical reconstructions of the Indian Ocean and Southeast Asia from 50–10 mya (A–F).

Adapted from Hall (2012). Light-grey represents shallow seas above continental shelves. Darker grey areas are subaerial regions on the continents. Base Map created using simplemappr.net.

Ancestor A1 diverged in the Himalayan region ∼43.8 mya, giving rise to A2—which remained in the Himalayan region—and A3. A3 dispersed to the Indochina region through mountainous corridors along the southeastern edge of an emerging Qinghai-Tibetan Plateau (Farve et al., 2015) as these basal divergences formed chronologically separate lineages in a west to east pattern from the Himalayas to Indoburma (Wood Jr et al., 2012; Agarwal et al., 2014). One of these divergences involving A2 in the Himalaya region, occurred ∼39.3 mya and resulted in the formation of the fasciolatus group and A4—the latter of which dispersed into the lowlands of the Indoburma region along north-south oriented hilly corridors (Farve et al., 2015; Ding et al., 2017) where it radiated ∼35.4 mya into the peguensis and khasiensis groups. This coincides with the intensifying orogeny in the region, which during the early Oligocene formed the main Indoburman ranges: Arakan, Chin, and Naga Hills (Brunnschweiler, 1966) and would have facilitated the invasion of new habitats and speciation from a habitat generalist ancestor (Grismer et al., 2020). The peguensis group began to radiate ∼33.2 mya from the southern foothills of the Himalayas to the eastern edge of the Ayeyarwady Basin (Grismer et al., 2021a: Fig. 24) and the khasiensis group began to radiate ∼29.5 mya from generally south of the Brahmaputra River and east of the Padma River to the eastern edge of Ayeyarwady Basin (Agarwal et al., 2014; Grismer et al., 2021a: Fig. 15). The biogeography of the species within the khasiensis group is discussed further in Agarwal et al. (2014). The fasciolatus, khasiensis, and peguensis groups were referred to as the Myanmar clade in Wood Jr et al. (2012).

Orogenic events in what is currently northern Indochina between ∼40 and 35 mya, may have precipitated sequential, cladogeneic events among the Indochinese ancestors A3, A6, A12, and A14. A3 diverged into two lineages within Indochina ∼40.9 mya. One lineage, A5, radiated in karstic habitats throughout central Indochina from approximately 33.1 mya to present, giving rise to the angularis group (Grismer et al., 2021a: Fig. 9). The other lineage A6, diverged ∼39.5 mya and gave rise to A12, which remained in Indochina, and A7 which dispersed into Sundaland—most likely into current-day northern Borneo as they were a contiguous landmass during this period (Hall, 2013; De Bruyn et al., 2014; Fig. 5; Cao et al., 2017). This portion of Sundaland remained subaerial throughout the tectonic evolution of Southeast Asia, was covered with perhumid rainforests since at least the middle Eocene, and became a major center of origin for a vast number of clades of plants and animals (De Bruyn et al., 2014; Grismer et al., 2016; Morley, 2018, and references therein), including Cyrtodactylus (see below).

As A7 began to diverge ∼33.1 mya in Sundaland it became the ultimate source of origin for many species and species groups throughout Sundaland, Wallacea, Papua, and the Philippines, and included one back-dispersal into Indochina. The cladogenic events within the descendants A7 were likely the result of the dynamic and rapidly fluctuating insular landscapes of Sundaland from ∼30–10 mya (Hall, 2013; Fig. 5). A7 gave rise to a clade comprised of the lateralis and sworderi sister groups which radiated in western Sundaland (Grismer et al., 2021a: Fig. 20) from ∼27.7 mya to present. Grismer & Davis (2018) and O’Connell et al. (2019) discuss the biogeography of the laterlais and sworderi groups. A7 eventually gave rise to three ancestors that independently dispersed into Wallacea. One, presumably made an over-water dispersal onto the western Sulawesi landmass sometime after ∼28.2 mya when it diverged from the malayanus group and is represented today by C. spinosus. The western part of Sulawesi was connected to the Sunda shelf until at least 25 mya (Cao et al., 2017). The malayanus group radiated in Borneo from ∼20.2 mya to present. Davis et al. (2020) discuss the biogeography of the malayanus group. Wood Jr et al. (2012) hypothesized that regional endemism in mainland Southeast Asia may have been facilitated by long-term geographic barriers and is further supported here with substantially more species sampling (see below).

Another descendent of A7, A8, dispersed into Wallacea ∼29.1 mya, giving rise to the darmandvillei group which began to radiate ∼23.4 mya. Members of the darmandvillei group dispersed over-water back and forth across what is currently recognized as Wallace’s Line into Sundaland (O’Connell et al., 2019) at least three times between ∼19.6–5.3 mya (Grismer et al., 2021a: Fig. 7). The colonization of northwestern Australia by C. kimberleyensis happened no earlier than 5.3 mya when it diverged from its undescribed sister species on Timor Island (Wood Jr et al., 2012). Cyrtodactylus sadleiri on Christmas Island diverged no earlier than 3.9 mya from its undescribed sister species on Bali Island, Indonesia. A third Sundaic descendent, A9, diverged ∼29.1 mya and gave rise to a clade comprised of the agamensis and marmoratus groups, likely as a result of intensifying volcanism and orogeny on the western edge of Sundaland (Hall, 2012). These groups separated in Sundaland ∼27.7 mya and the agamensis group radiated in western Sundaland from approximately 24.3 mya to present (Grismer et al., 2021a: Fig. 7), whereas the marmoratus group began radiating in southern Sundaland ∼23.7 mya and dispersed to the Papua region through Wallacea (Grismer et al., 2021a: Fig. 27) ∼10.4 mya. Grismer & Davis (2018) and O’Connell et al. (2019) discuss the biogeography of the agamensis group.

As sequential cladogenic events—first initiated in A7—continued throughout Sundaland, most likely in what is current-day Borneo, one lineage ∼29.9 mya, gave rise to ancestor A10 of the philippinicus group which sequentially gave rise to two Bornean clades (Wood Jr et al., 2012). The first at ∼26.7 mya, is currently represented in part by the Cyrtodactylus pubisulcus species complex (Davis et al., 2020). The second Bornean clade, represented today by at least 10 species, diverged ∼24.3 mya and radiated in Borneo while its sister lineage dispersed to the Philippines sometime afterwards. The biogeography of the species of both Bornean clades are discussed by Davis et al. (2020). The Philippine clade began radiating in situ ∼23.1 mya, indicating its route to this archipelago was along a southern chain of islands onto Mindanao as it was the only available route during this time period (Hall, 2013; Fig. 5). Siler et al. (2010), Siler et al. (2012), Welton et al. (2010a) and Welton et al. (2010b) discuss the biogeography of the Philippine gekkonids and other possible routes. Eventually, A7 gave rise to an Indochinese descendent A11 ∼30.1 mya, which dispersed back into the Indochina region no later than ∼27.5 mya during a time period when broad, subaerial upland areas still linked northern Borneo and southern Indochina across an exposed Sunda Shelf (Figs. 5D–5E). A11 diverged, giving rise to a clade of at least 33 species comprising the condorensis and irregularis groups. The condorensis group is centered in southern Vietnam and radiated from ∼23.7 mya to present and the irregularis group radiated in Vietnam, Laos, and Cambodia from ∼23.3 mya to present (Grismer et al., 2021a: Fig. 9). One species of the condorensis group, C. leegrismeri, occurs on two small islands off the east coast of the Burma-Thai-Malay Peninsula in Sundaland. Grismer & Grismer (2017) discuss the biogeography of the condorensis group and Nguyen et al. (2017) discuss the biogeography of the irregularis group.

Approximately 38.1 mya, ancestor A12 diverged in Indochina, giving rise to the Indochinese ancestor A14 and the ancestor of the triedrus group (A13) which dispersed to the India-Sri Lanka region at a time when the Indian subcontinent was adjacent to Indoburma and Indochina (Acton, 1999; Köhler & Glaubrecht, 2007; Aitchison, Ali & Davis, 2007; Ali & Aitchison, 2008: Fig. 5). This invasion route to Indian-Sri Lanka from what was probably southern Indochina (being there is no evidence of this group ever occupying Indoburma), via a relatively narrow over-water dispersal or land bridge, has been hypothesized for a number taxonomic groups (e.g., (Klaus et al., 2010) [crabs]; (Li et al., 2013) [rhacophorid frogs]; J. (Grismer et al., 2016) [draconine lizards]; (Garg & Biju, 2019) [microhylid frogs]; (Gorin et al., 2020) [microhylid frogs]). Given the extensively greater sampling of Cyrtodactylus here and the increased use of tectonic data, this scenario does not support long distance over-water dispersal scenarios across the Bay of Bengal proposed by Wood Jr et al. (2012) and Agarwal et al. (2014) nor does it support the scenario of Grismer & Davis (2018) that Sundaland was colonized by an ancestor of the Indian subcontinent. Based on the above interpretations, the triedrus group began diversifying in mesic areas no later than ∼33.3 mya and has become generally restricted to the periphery of the subcontinent south of the Ganga River following gradual aridification during the Oligocene (Morley, 2018; Deepak & Karanth, 2018). Agarwal & Karanth (2015) discuss the biogeography of the terrestrial members of this group.

A14 remained in Indochina and continued to radiate, eventually diverging ∼35.8 mya and giving rise to A19, which also remained in Indochina, and A15 which invaded the Papua region. This colonization event is problematic. Unlike the invasion of the Papua region by the marmortus group which left a chronological sequential trail of ancestors throughout Sundaland and Wallacea, no such trail exists for A15. From ∼40–30 mya, a series of islands that later coalesced to form Sulawesi existed to the east of southern Indochina—which at the time was contiguous with current-day Borneo—and could have provided a route into the Papua region (Hall, 2013; Fig. 5). The absence of closely related species in current-day Sundaland and Wallacea neither supports nor discounts this hypothesis. A15 diverged ∼32.7 mya in the Papua region and gave rise to A16—the ancestor of a monophyletic Melanesia radiation that currently consists of at least 36 species comprising seven monophyletic species groups that radiated throughout the Papua region (including northeastern Queensland) from ∼29.5 mya to present (Grismer et al., 2021a: Figs. 37, 39). Oliver, Skipwith & Lee (2014) and Oliver et al. (2018) discuss the complex biogeographic history of this clade. At 32.7 mya, A15 also gave rise to A17 in Papua New Guinea which diverged ∼28.2 mya and may have given rise to Cyrtodactylus biordinis in the Solomon Islands (Oliver et al., 2018). The putative sister lineage of C. biordinis, A18, dispersed to and radiated on the Burma-Thai-Malay Peninsula in western Sundaland south of the Isthmus of Kra as the pulchellus group (Grismer et al., 2021a: Fig. 16) from ∼18.3 mya to present. Like the dispersal from the Indochina region to the Papua region, this too is a difficult scenario to explain given that there are no closely related, earlier diverging species in the intervening regions of current-day Wallacea or in southern Sundaland. Grismer et al. (2014) discuss the biogeography of the pulchellus group.

A19, a direct descendent of A14 in the Indochina region, is the ancestor of a large Indochinese clade containing at least 78 species that comprises eight species groups. Grismer & Davis (2018) discuss the complex tectonic history that this vast, corrugated landscape of intermontane rift basins and parallel mountain ranges (Feton et al., 1997; Fenton, Charusiri & Wood, 2003) played in the diversification of this clade from eastern Myanmar to central Thailand and others (Chen et al., 2017; Chen et al., 2018; Grismer et al., 2018; Poyarkov et al., 2018; Suwannapoom et al., 2018; Gorin et al., 2020), as the Tenasserim Mountains and Thai Highlands were gradually uplifted (Upton, Bristow & Hurford, 1995; Upton et al., 1997). As A19 and its descendants continued to diverge and radiate in Indochina, the first ancestors to evolve were A20 and A21, ∼34.2 mya. A21 diverged approximately 29.7 mya, giving rise to the brevipalmatus group centered in Indochina and the linnwayensis group of the Shan Plateau of eastern Myanmar (Grismer et al., 2021a: Figs. 9, 24). Approximately 21.0 mya, the breviplamatus group dispersed across, what is currently the Isthmus of Kra on the Burma-Thai-Malay Peninsula, into northern Sundaland to as far south as Peninsular Malaysia where it is currently represented by C. elok and C. breviplamatus.

Ancestor A20 continued to radiate in the Indochina region and at ∼32.9 mya, it diverged, giving rise to A24 of the intermedius group which then radiated throughout eastern Thailand, Cambodia, and southern Vietnam from ∼24.6 mya to present (Grismer et al., 2021a: Fig. 16). A20 also gave rise to ancestor A22—a clade of Indochinese species comprised of the sadansinensis, yathepyanensis, oldhami, sinyineensis, and chauquangensis groups. The earliest diverging lineages of that clade are the sadansinensis (∼30.1 mya) and the yathepyanensis (∼28.0 mya) groups from the Salween Basin of southeastern Myanmar (Grismer et al., 2021a: Fig. 24). The next group to diverge, the oldhami group (∼26.7 mya), is endemic to the Burma-Thai-Malay Peninsula and contains one species, C. zebraicus, that diverged in northern Sundaland ∼13.4 mya, just south of the Isthmus of Kra (Grismer et al., 2021a: Fig. 16) and another undescribed Sundaic species from Krabi, Thailand, that is closely related to C. sanook, which also diverged south of the isthmus ∼5.5 mya. The most recently diverged sister lineages, the sinyineensis and chauquangensis groups, radiated throughout Indochina in allopatry with respect to one another. The sinyineensis group is a disjunctly distributed, saxicolous lineage that began radiating ∼22.3 mya and extends from the Salween Basin of southeastern Myanmar through the mountainous terrain of western Thailand to at least Doi Inthanon (Grismer et al., 2021a: Fig. 24). The chauquangensis group radiated from ∼21.7 mya to present, across karstic landscapes in northern Vietnam, southern China, Laos, and northern Thailand (Grismer et al., 2021a: Fig. 9).

Given that the mtDNA phylogeny here is highly congruent with the mito-nuclear phylogeny of Grismer et al. (2021a), we suspect that adding more species and genes will have little bearing on the phylogeny and the biographical scenario presented here. However, we are in the process of constructing a genomic data set with additional species that will be compared to the mitochondrial and mito-nuclear data sets of the most recent genus-wide phylogenies of Grismer et al. (2021a) and Grismer et al. (2021b).

Conclusions

The dispersal and colonization capabilities of the hyper-diverse gekkonid genus Cyrtodactylus are not greatly influenced by well-established biogeographic barriers as evidenced by their ability to cross seaways, major river systems, basins, mountain ranges, and ephemeral land bridges—followed by extensive in situ diversification within specific geographic areas and habitats. Cyrtodactylus originated in a proto-Himalayan landscape during the early Eocene and later dispersed into and radiated in Indoburma and Indochina. The Indoburma radiation remained in situ but the Indochina radiation became a major center of origin that since the Late Eocene, seeded the evolution of the remainder of the genus with three independent dispersal events and subsequent radiations in Sundaland, India-Sri Lanka, and Papua. The Sundaland radiation became a second major center of radiation during the Oligocene with dispersals and subsequent dispersals and radiations in Wallacea, the Philippines, and back to Indochina. A Miocene dispersal from Indochina to Papua and another from Papua to Sundaland are difficult to explain given that the intervening geographic region (Wallacea) existed only as a series of small islands. The discovery of related species in Wallacea would add further support to what at this point can only be hypotheses of over-water dispersal.

Supplemental Information

Table S1 Species, habitat preference with supporting references, species group designations, and GenBank accession numbers for specimens used in the phylogenetic analysis

Species can be cross-referenced to Figure 3 by GenBank accession.

Click here for additional data file.

Supplemental Information 1 Cyrtodactylus geofile

Click here for additional data file.

Supplemental Information 2 Cyrtodactylus BEAST .tre file

Click here for additional data file.

Supplemental Information 3 Cyrtodactylus nexus file

Click here for additional data file.

PLW’s collaboration on this paper constitutes contribution number 946 of the Auburn University Museum of Natural History. NA Poyarkov thanks Andrei N. Kuznetsov (JRVTC, Hanoi, Vietnam) and Leonid P. Korzoun (MSU, Moscow, Russia) for support of his work in Indochina, and is grateful to Chatmongkon Suwannapoom and Parinya Pawangkhanant (AUP, Thailand), Platon V. Yushchenko, Anna S. Dubrovskaya, Sabira S. Idiatullina, and Vladislav A. Gorin (MSU, Russia) for various support and assistance in the field and in the lab. We that Todd E. Jackman and Matthew P. Heinicke more helpful reviewer comments on the manuscript.

Additional Information and Declarations

Competing Interests

Author Contributions

Data Availability

Nikolay A. Poyarkov is an Academic Editor for PeerJ. The other authors declare that they have no competing interests.

L. Lee Grismer, Nikolay A. Poyarkov, Jesse L. Grismer and Perry L. Wood Jr conceived and designed the experiments, performed the experiments, analyzed the data, prepared figures and/or tables, authored or reviewed drafts of the paper, and approved the final draft.

Evan S.H. Quah conceived and designed the experiments, performed the experiments, analyzed the data, authored or reviewed drafts of the paper, and approved the final draft.

The following information was supplied regarding data availability:

This study is based on phylogenetic analysis of publicly available DNA sequences at GenBank (Table S1).

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
