# Peer review of "The biogeography of bent-toed geckos, Cyrtodactylus (Squamata: Gekkonidae)"

_PeerJ, doi:10.7717/peerj.13153_

## Round 0.1 · original submission · Minor Revisions

Both reviewers have provided very thorough and thoughtful reviews that should help you with a revision. As you will see, the reviewers have different opinions on your manuscript, and there are a number of points that need attention. Although Rev#1 has listed some substantial revisions, many of these are suggested and as he points out will make for a much better publication, but are not strictly speaking necessary for acceptance in PeerJ. Therefore, I will draw attention to revisions that are required and leave it up to the authors to decide whether or not they wish to conduct the substantial changes and reanalyses suggested by Rev#1.

1. You need to update (Genbank numbers & species names) and thoroughly describe the supplemental information (SI) for this manuscript. Nexus, tree files, etc. must be deposited either with this SI, or a link provided to another repository.

2. All relevant literature must be cited and comparisons made where necessary (e.g. Xu et al. 2020). It would certainly be more useful if you could point out whether trees are congruent, and/or where differences occur (also mentioned by Rev#2).

3. Rev#1 is correct in that you should look forward to what more complete analyses might show and where your own data fall short.

I look forward to seeing your rebuttal and revision.

·

Basic reporting

The figures are excellent and the reconstructions of biogeographic history are also very good. There is an important missing reference - Xu et al. 2020 - see my comments further in the article.

Experimental design

Most of my review will be contained in the 'Validity of the findings' section - In terms of experimental design, the paper lacks explicit comparisons to past work with the purpose of providing a hypothesis that will likely be robust to future additions and improvements in sampling and methodologies.

Validity of the findings

In my opinion, in order to publish a paper with no new data, the bar for analyses and results have to be higher in order to justify the publication. This bar was not met for this paper for a number of reasons. I think that the paper is worthy of publication, but only after addressing a number of issues and improving the paper substantially.
Here is an overview of the issues needed to make the paper acceptable for publication:

1. The biggest thing missing from this paper are quantitative and qualitative comparisons to previous work which, if completed, would allow a prediction about how robust the biogeographic conclusion are and how sensitive the conclusions are to 1. improvements in sampling 2. improvements/changes in methodology of reconstruction 3. Increased knowledge surrounding the nuclear genes and past introgression events.
By making these comparisons and addressing the issue of repeatability, this paper could be the new standard for Cyrtodactylus biogeography. Without these comparisons, it is not clear that this paper provides any unique insights into the biogeography of this group. If I had to guess, I would say that this paper is congruent with other biogeographic interpretations of the group and will be robust to the addition of taxa and improvements in methodologies in the future, but concluding that requires careful comparisons.


2. The supplemental data is not sufficient and needs substantial editing and additions. A more complete description of the data both the text of the manuscript and in the supplemental data are needed.

Details regarding the above 2 points:
1. The following needs to be done in order to assess how different and robust this biogeographic hypothesis is:
First, to what degree is the biogeographic hypothesis in this paper different from Wood et al. 2012 and from Xu et al. 2020?
Xu et al., was not cited in the manuscript, but is important to consider - Xu, W., Dong, W.J., Fu, T.T., Gao, W., Lu, C.Q., Yan, F., Wu, Y.H., Jiang, K., Jin, J.Q., Chen, H.M. and Zhang, Y.P., 2021. Herpetological phylogeographic analyses support a Miocene focal point of Himalayan uplift and biological diversification. National science review, 8(9), p.nwaa263.
Xu et al. contains biogeographic hypotheses for both the Himalayas by itself, and an analysis that includes more categories - 6 areas for Cyrtodactylus. See the supplemental data for tree graphics and the trees with branch lengths included. Non-Cyrtodactylus geckos could be pruned out for these comparisons.
Are the trees the same? Because sampling is so complete, it would be possible to trim the number of taxa down to exactly match Wood et al. and Xu et al.
Are the topologies the same? If not, use Bayes factors to estimate how much they differ. How different are the analyses used to estimate the trees in all three cases?

Given a careful comparison of the topologies, if there are any differences in biogeographic reconstructions, do the differences in topology account for those differences? If the biogeographic hypotheses are the same, then it suggests that it may be robust to future additions of taxa and improvements in methodology.

My guess is that all three of these have congruent topologies and congruent biogeographic hypotheses. It is possible that the only difference is that the current manuscript is better resolved and included more areas than Xu et al. A quantitative proof of that guess would help to improve the importance of this paper.

Point 2 above: The supplemental data is the supplemental data from the Grismer et al. 2021 Diversity paper unchanged. The references in the supplemental table are relevant for that Diversity paper, but not for this one. The references need to be replaced with the sources for the Genbank entries for each species, Wood et al. 2012, Griser et al., etc.
For some of the species, the number entered is not a Genbank number, but is a specimen number.
Here are those entries with no genbank numbers:
C. cf. brevipalmatus arboreal herein brevipalmatus USMHC 2555
C. tibetanus general K. Wang. Personal communication, 2019 tibetanus YPX1413
C. zhaoermi general Shi and Zhao [144] tibetanus YPX1433

The Xu et al. paper has genbank numbers for the C. tibetanus, and C. zhaoermi

The supplemental data should include the nexus file with all of the sequence data for the analysis in this paper, along with the tree files with branch lengths for all of the results.

The sequence data in this paper is presented as being ND2 plus 5 tRNAs, but the sequence data presented falls into 2 categories, some of this variation in sequence completeness needs to be addressed. I don't think it will make much difference in the end, but that should be proven in the paper by either excluding incomplete sequences and showing that the results are not different, or be making an argument that the incomplete sequences are not in crucial places in the tree, or both.
Category 1, complete - Category 2, ND2 only (or less) -

If this data set is going to be reanalyzed, there are a number of newly described Indian taxa that are distinct enough to merit inclusion including - C. aaronbaueri, C. agarwali, C. bapme, and C. bengkhuaiai

The other issue that needs to be discussed if the biogeographic hypothesis is meant to be robust and long lasting is the nuclear data. Is the nuclear data presented in Wood et al. sufficient to predict future congruence of mitochondrial and nuclear data? I think a caveat needs to be made that future nuclear sequence data may prove some of the biogeographic hypotheses incorrect. For example, in Anolis, the overall biogeographic hypotheses are supported by mtDNA and are congruent with nuclear data - the anolis genome paper analyzed 96 loci and found it to be congruent with mtDNA. However, within Jamaican anoles, it was recently shown (Meyers et al., 2021 in Sys. Bio) that mitochondrial introgression has occurred making some of the previous hypotheses based only on mtDNA incorrect.

Additional comments

I have withheld from specific comments on the manuscript because the revisions are major enough, that an acceptable version would have many changes.

To summarize what would make this paper acceptable:
1) Comparisons to previous hypotheses, including Bayes factors comparing different topologies and a discussion of differences in biogeographic hypotheses.
2) An updated, revised and expanded supplemental data that includes the all the sequences, branch lengths, and appropriate references.
3) A discussion that includes speculation about the future utility of the biogeographic hypotheses presented in this paper, including the possibility of conflict or congruence of nuclear genes.

·

Basic reporting

The manuscript is very well written, cites the previous literature extensively, and is structured appropriately for a study focused on presenting the results of a phylogenetic comparative method. I have only the following minor suggestions:

-there are a few places where commas are used that seem unneccessary: lines 21, 47, 248, 275, 350, 426

-the statement on lines 121-122 on region boundaries should include citations, or else state something like "see below for details" since the boundaries are discussed with citations in subsequent paragraphs.

-on line 257 "A" for the node IDs is in parentheses but that format isn't used elsewhere

-on line 306 Timor-Leste is used but from context it seems like the island of Timor is being referred to rather than the country

-there seems to be a word or words missing after "A11 diverged" on line 311. I think it is supposed to be a date or else region.

-some of the text in Figs 2 and 3, including species names, is not legible in the reviewer manuscript. This might just be because reviewers are sent lower resolution figures; if so, ignore.

Experimental design

Experimental design is straightforward. The study uses a previously published phylogeny and a new biogeographic analysis which are appropriate for the study goals. I have only a few minor comments:

-On line 123, it is noted that each species is assigned to a single geographic area. Are there any species that cross the boundaries used in this study? If there are, which species were affected and how was the "home" region defined? If not, perhaps rephrase to indicate that no species occur across the region boundaries.

-I realize the tree was already published, but it would be useful to expand slightly in a couple spots. First, was the 350 million generation length of the run just what was needed to reach sufficient ESS values, or does it have some other significance? Second, on line 204 it is noted that the backbone of the tree matches multilocus data sets, justifying use of ND2 alone. It would be worth citing the relevant studies being compared here that employed multilocus data.

Validity of the findings

In general, the conclusions are straightforward and describe the patterns observed in the timing and biogeographic analyses. I have only a few suggestions:

-Line 211 states that the best model is DEC but the table shows it to be DEC + J, and that is the model shown in Fig. 3.

-on Line 246 it is noted that the discussion uses mean dates for cladogenic events. It would be worth stating here how broad a typical HPD range is just to provide a sense without having to refer to Fig. 3 or clutter the subsequent text.

-The introduction frames this study as an update of previous studies using a much denser phylogeny and more detailed biogeographic reconstructions. As such, it would be worth adding one sentence to the discussion describing the overall degree of agreement between these studies and the conclusions drawn here, with the caveat that only some patterns are directly comparable across studies. For example, there is one major difference with the Wood et al paper pointed out pertaining to the triedrus group, but similarities are generally not noted so a blanket statement would usefully cover this.

-Lines 338-349 posit a overland or short overwater dispersal from Indochina to the Indian Subcontinent to explain the distribution of the triedrus group. Fig. 5 shows the Indian subcontinent as adjacent to Indoburma, not Indochina, with a midsized shallow-water gap in the 35-40 million year time frame. Do the authors think that extinction in the intervening Indoburma region is a better explanation than a longer overwater dispersal that bypassed Indoburma?

-On Lines 351-353 it should be noted that the triedrus group pattern is the authors' interpretation based on the overland dispersal posited in the previous lines since there isn't direct evidence that the group formerly occupied the northern Indian subcontinent.

Additional comments

Very nice study overall. Although certainly not necessary, it would be great if some of the biological patterns were superimposed on the paleogeographic reconstructions included in Fig. 5. For example, regions inferred to be occupied by Cyrtodactylus at each time point could be more intensely colored, or arrows added to indicate dispersals similar to those used in Fig. 1.

---

## Round 0.2 · accepted · Accept

In making this decision to accept the current version, I do so with some reluctance. Although the authors have responded to the comments of the reviewers, my reading is that they have not responded in the positive spirit of peer review. Moreover, responses to my own comments are absent.

Changes to the manuscript are negligible (and negligent - with respect to spelling and punctuation).

Given that the authors have been given several opportunities to improve their rebuttal and have opted not to, I have been left with an unenviable decision. I would urge them to carefully read their manuscript and correct the outstanding grammatical and spelling errors. It would also be of benefit to their readers if they would spend more time reflecting on the purpose of the review process.